# Prognostic Impacts of LL-37 in Relation to Lipid Profiles of Patients with Myocardial Infarction: A Prospective Cohort Study

**DOI:** 10.3390/biom12101482

**Published:** 2022-10-14

**Authors:** Runzhen Chen, Hanjun Zhao, Jinying Zhou, Ying Wang, Jiannan Li, Xiaoxiao Zhao, Nan Li, Chen Liu, Peng Zhou, Yi Chen, Li Song, Hongbing Yan

**Affiliations:** 1Department of Cardiology, Fuwai Hospital, National Center for Cardiovascular Diseases, Peking Union Medical College and Chinese Academy of Medical Sciences, Beijing 100037, China; 2Fuwai Hospital, Chinese Academy of Medical Sciences, Shenzhen 510000, China; 3Coronary Heart Disease Center, Fuwai Hospital, Chinese Academy of Medical Sciences, Beijing 100037, China

**Keywords:** LL-37, myocardial infarction, lipoprotein(a), proprotein convertase subtilisin/kexin type 9

## Abstract

*Background.* In vivo studies show that LL-37 inhibits the progression of atherosclerosis and predicts a lower risk of recurrent ischemia in patients with acute myocardial infarction (AMI), which could be mediated by the modulation of lipid metabolism. The current study aimed to investigate the effects of various lipid contents on the prognostic impacts of LL-37 in patients with AMI. *Methods.* A total of 1567 consecutive AMI patients were prospectively recruited from March 2017 to January 2020. Patients were firstly stratified into two groups by the median level of LL-37 and then stratified by levels of various lipid contents and proprotein convertase subtilisin/kexin type 9 (PCSK9). Cox regression with multiple adjustments was performed to analyze associations between LL-37, lipid profiles, PCSK9, and various outcomes. The primary outcome was major adverse cardiovascular event (MACE), a composite of all-cause death, recurrent MI, and ischemic stroke. *Results.* During a median follow-up of 786 (726–1107) days, a total of 252 MACEs occurred. A high level of LL-37 was associated with lower risk of MACE in patients with elevated lipoprotein(a) (≥300 mg/L, hazard ratio (HR): 0.49, 95% confidence interval (CI): 0.29–0.86, *p* = 0.012) or PCSK9 levels above the median (≥47.4 ng/mL, HR: 0.57, 95% CI: 0.39–0.82, *p* < 0.001), which was not observed for those without elevated lp(a) (<300 mg/L, HR: 0.96, 95% CI: 0.70–1.31, *p* = 0.781, *p*
_interaction_ = 0.035) or PCSK9 (<47.4 ng/mL, HR: 1.02, 95% CI: 0.68–1.54, *p* = 0.905, *p*
_interaction_ = 0.032). *Conclusions.* For patients with AMI, a high level of LL-37 was associated with lower ischemic risk among patients with elevated lp(a) and PCSK9.

## 1. Introduction

Dyslipidemia, especially hypercholesterolemia, is one of the most important risk factors for the development and progression of coronary artery disease (CAD) [1,2]. According to the latest guidelines, a low-density lipoprotein cholesterol (LDL-C) level of 1.4–1.8 mmol/L or lower is recommended for patients with myocardial infarction (MI) to achieve with the routine use of lipid-lowering medications [3]. Although statins demonstrate high efficacy in lowering LDL-C, recent studies report that the use of statins could increase circulating levels of lipoprotein(a) (lp(a)), an increasingly important risk factor for atherosclerosis [4,5]. Meanwhile, over 40% of patients fail to achieve LDL-C targets despite statin medications of high intensity, and many patients still suffer from residual ischemic risk caused by persistently high levels of LDL-C [6]. Consequently, non-statin lipid-lowering medications have been developed and applied to patients irresponsive to traditional statin treatments, including ezetimibe and proprotein convertase subtilisin/kexin type 9 (PCSK9) inhibitors, which only partially solves the issue, as the former only brings a mild reduction in cholesterol, while the latter suffers from inconveniences due to injections and extra medical costs [2,3]. Therefore, the continuous search for new agents to tackle residual lipid risk is still of clinical interest, in order to achieve better control of the whole lipid profile with appropriate cost-effectiveness.

LL-37, an antimicrobial peptide mainly produced by neutrophils, has been identified in atherosclerotic plaques and shown to be involved in the development of atherosclerosis [7,8]. Previously, we have reported, in a small-scale prospective cohort, that a high level of LL-37 is associated with a lower risk of ischemic events for patients with ST-segment elevation myocardial infarction (STEMI) treated by percutaneous coronary intervention (PCI), but the mechanisms behind it are still undetermined [9]. In fact, the majority of LL-37 is bound to various lipoproteins within the circulation [10,11]. Earlier studies also show that LL-37 reduces the expression of CD36 and inhibits lipid accumulation in adipocytes and hepatocytes, while statin medications tend to increase LL-37 levels [12,13]. These findings suggest possible interactions between LL-37 and various lipid contents, but relevant reports in an atherosclerotic population are quite limited. Therefore, this study aimed to investigate the prognostic impacts of LL-37 in relation to the lipid profiles of patients with acute myocardial infarction (AMI), in order to offer more insights regarding the interplay between LL-37 and lipid risk.

## 2. Materials and Methods

### 2.1. Study Cohort

This prospective cohort study was conducted in a large-volume national tertiary care institute (Fuwai Hospital, Beijing, China) specializing in cardiovascular diseases, which has consecutively enrolled patients (age ≥ 18 years) who underwent emergent coronary angiography (CAG) due to the diagnosis of AMI (symptom onset ≤ 24 h before presentation) from March 2017 to January 2020. The diagnosis and classification of AMI were made according to up-to-date guidelines and universal definitions [3,14], including criteria of clinical presentations, typical characteristics from electrocardiography, dynamic changes in cardiac enzymes, and imaging evidence. The current cohort had recruited all patients diagnosed with STEMI and non-ST-elevation MI (NSTEMI), who were later triaged to conservative medical treatment, PCI, or coronary bypass grafting (CABG) based on results of CAG and clinical conditions. Patients were excluded from the final analysis if they had no available measurements of plasma LL-37, PCSK9, complete lipid profiles (including LDL-C, high-density lipoprotein cholesterol (HDL-C), lp(a), and triglyceride), or no records of follow-up. The study was performed in accordance with principles set forth in the Declaration of Helsinki, was approved by the ethics committee of the institute (No. 2017-866), and all patients signed written informed consent during hospitalization.

### 2.2. Blood Samples Collection and Measurements

Blood samples for complete blood count, basic metabolic panel (e.g., creatinine, glucose), cardiac troponin I (cTnI), and N-terminal prohormone of brain natriuretic peptide (NT-proBNP) were immediately collected via cubital vein right after patient admission to the hospital. Blood cell count was carried out using an automatic hematology analyzer (XT-1800i; Sysmex Corporation). Concentrations of blood glucose and creatinine were measured with an automatic biochemistry analyzer (Hitachi 7150, Tokyo, Japan) following standard protocols set up by the core laboratory. Serum cTnI was determined using the immunochemiluminometric assay (Access AccuTnI, Beckman Coulter, Brea, CA, USA). Plasma levels of NT-proBNP were measured with electro-chemiluminescent immunoassays (Elecsys proBNP II assay, Roche Diagnostics, Mannheim, Germany).

Blood samples for measurements of LL-37 and PCSK9 were collected via radial or femoral access before the initiation of CAG using vacutainer tubes containing EDTA, which were immediately centrifuged at 2000× *g* for 15 min at room temperature for the isolation of plasma, and then stored at −80 °C until further analysis. Enzyme-linked immunosorbent assays were used for measurements of plasma LL-37 (HK321, HyCult Biotechnology, Uden, Netherlands) and PCSK9 (DY3888, R&D Systems, Minneapolis, MN, USA), which were performed according to the manufacturer’s instructions, respectively.

Blood samples for lipid profiles and other routine tests were collected via the cubital vein right after the patients were admitted into the coronary care unit. The plasma concentrations of triglyceride, LDL-C, and HDL-C were measured with an automatic biochemistry analyzer (Hitachi 7150, Tokyo, Japan), while serum levels of lp(a) were determined by the immunoturbidimetry method (LASAY Lp(a) auto, SHIMA Laboratories Co., Ltd., Tokyo, Japan). HbA1c levels were measured using Tosoh Automated Glycohemoglobin Analyzer (HLC-723G8, Tokyo, Japan). The level of high-sensitivity C-reactive protein was measured using immunoturbidimetry (Beckmann Assay, Bera, CA, USA).

### 2.3. Outcomes and Follow-Up

The primary outcome was major adverse cardiovascular event (MACE), defined as a composite of all-cause death, recurrent MI, and ischemic stroke. Secondary outcomes included each component of the primary outcome, cardiac death, and a composite endpoint of cardiac death, recurrent MI, and ischemic stroke. Patients were routinely followed up after discharge. For the 1st month of follow-up, patients were asked to come back to the institute for comprehensive examinations. For those who could not come to the institute, the follow-up was completed independently by staff from the information center of the institute using a standardized questionnaire through phone-call interviews. After that, all patients would be interviewed again at the 6th and 12th months through phone calls. For those who survived more than a year, the subsequent follow-up would be made annually. The outcome data were transferred to the research group on a monthly basis, and a group of physicians routinely assessed the reported adverse events.

### 2.4. Statistical Analysis

All statistical analyses were performed using Stata 17.0 (StataCorp, College Station, TX, USA). Patients were firstly stratified into two groups (high vs low) by the median level of LL-37, then stratified by levels of LDL-C (≥2.6 mmol/L vs. <2.6 mmol/L), HDL-C (≥1 mmol/L vs. <1 mmol/L), lp(a) (≥300 mg/L vs. <300 mg/L), triglyceride (≥1.7 mmol/L vs. <1.7 mmol/L), and PCSK9 (stratified by the median, ≥47.4 ng/mL vs. <47.4 ng/mL). Univariable Cox regression was used to analyze associations between LL-37 levels, all other baseline variables, and various clinical outcomes. Subsequently, multivariable analyses for LL-37 and outcomes were performed with adjustments for variables with *p* < 0.1 identified in the univariable analysis (Appendix A) or all collected baseline variables. Categorial variables are presented as numbers (%) and analyzed with chi-squared tests. Continuous variables are presented using mean ± SD if they follow the normal distribution and are tested with a *t*-test. Otherwise, they are presented as medians with the 25th and 75th percentiles and tested by the Wilcoxon rank-sum test. The correlations between LL-37 and various lipid parameters or PCSK9 were calculated using the Spearman correlation coefficient. A two-tailed *p*-value < 0.05 was considered statistically significant.

## 3. Results

### 3.1. Patient Cohort and Baseline Characteristics

From March 2017 to January 2020, a total of 1723 patients underwent emergent CAG due to AMI at the institute. Among these patients, 140 patients did not have available LL-37 or PCSK9 measurements, 15 patients did not have complete lipid profiles, and 1 patient did not have any follow-up records. Finally, a total of 1567 patients were included in the current analysis (Figure 1).

Overall, the median level of LL-37 was 21.86 (13.89–32.67) ng/mL, the average age of patients was 60.7 ± 12.3 years, and most of the patients were male (80.6%) and presented with STEMI (88.6%). Compared to patients with low levels of LL-37 (Table 1), patients with high levels of LL-37 tended to have younger age, male sex, higher body mass index, and a history of smoking. Hemodynamics were similar between the two groups, except for a slight difference in diastolic blood pressure at admission. Regarding lab test results, patients with high LL-37 presented with higher levels of hemoglobin, leucocytes, platelets, high-sensitivity C-reactive protein, LDL-C, triglyceride, and PCSK9, but lower levels of N-terminal prohormone of brain natriuretic peptide and HDL-C. Furthermore, the distribution of culprit lesions, number of diseased vessels, results of revascularizations, and medications were similar between the two groups. According to Spearman correlation analysis (Figure 2), plasma levels of LL-37 were positively correlated with levels of LDL-C (r = 0.089, *p* < 0.001), triglyceride (r = 0.194, *p* < 0.001), and PCSK9 (r = 0.222, *p* < 0.001), but negatively correlated to levels of HDL-C (r = −0.177, *p* < 0.001), while its correlation with lp(a) was not statistically significant (r = 0.004, *p* = 0.869).

### 3.2. Associations between LL-37 Levels and Clinical Outcomes Stratified by Lipid Profiles

During a median follow-up of 786 (726–1107) days, a total of 252 (16.1%, incidence rate (IR): 63.0/1000-person-year (PY)) MACEs occurred, including 113 (7.2%, IR: 28.3/1000-PY) cases of all-cause death, 63 (4.0%, IR: 15.8/1000-PY) cases of cardiac death, 88 (5.6%, IR: 22.0/1000-PY) cases of recurrent MI, and 67 (4.3%, IR: 16.8/1000-PY) cases of ischemic stroke. According to univariable Cox regression, a high level of LL-37 was associated with a lower risk of MACE (hazard ratio (HR): 0.71, 95% confidence interval (CI): 0.55–0.91, *p* = 0.007), but its statistical significance was lost (HR: 0.68, 95% CI: 0.41–1.14, *p* = 0.141) after multiple adjustments for other baseline variables with *p* < 0.1 in the univariable analysis (Appendix A).

A stratified analysis showed that risk reduction associated with LL-37 was substantially affected by levels of lp(a) and PCSK9 (Figure 3 and Figure 4). High LL-37 was associated with a lower risk of MACE in patients with lp(a) ≥ 300 mg/L (HR: 0.49, 95% CI: 0.29–0.86, *p* = 0.012), which was not observed for those with lp(a) within a normal range (HR: 0.96, 95% CI: 0.70–1.31, *p* = 0.781, *p*
_interaction_ = 0.035). Significant interactions were also detected between LL-37 and PCSK9, as high LL-37 was associated with lower risk of MACE in patients with PCSK9 levels above the median (≥47.4 ng/mL, HR: 0.57, 95% CI: 0.39–0.82, *p* < 0.001), but not in those with lower PCSK9 (<47.4 ng/mL, HR: 1.02, 95% CI: 0.68–1.54, *p* = 0.905, *p* _interaction_ = 0.032). On the other hand, associations between LL-37 and MACEs were generally not affected by variations of other lipid parameters, as indicated by consistent HR values without statistical significance in stratified analysis. Similar risk reduction and interactions between LL-37 with lp(a) or PCSK9 were observed with adjustments for all collected baseline variables (Appendix A).

To determine the mutual independence of the interactions between LL-37 and lp(a) and PCSK9, we simultaneously included two interaction terms (LL-37#lp(a) and LL-37# PCSK9) into the multivariable regression models (Appendix A), and both interaction terms remained statistically significant (both *p* < 0.05). In a stratified analysis of combinations of lp(a) and PCSK9 (Appendix A), high levels of LL-37 were associated with a trend towards a lower risk of MACE among patients with either elevated PCSK9 (HR: 0.65, 95% CI: 0.41–1.02, *p* = 0.062) or lp(a) (HR: 0.42, 95% CI: 0.16–1.13, *p* = 0.085), while high LL-37 was associated with the greatest and most significant reduction in MACE risk in patients with both parameters elevated (HR: 0.45, 95% CI: 0.21–0.95, *p* = 0.036). Significant interactions were detected between LL-37 and combinations of lp(a)/PCSK9 (*p*_interaction_ = 0.030).

Differentiations in risk reduction associated with LL-37 were also observed for secondary outcomes (Appendix A). For the composite endpoint of cardiac death, recurrent MI, and ischemic stroke (Figure 5), high LL-37 was still associated with substantially lower risk for patients with elevated lp(a) (HR: 0.49, 95% CI: 0.28–0.85, *p* = 0.012) or PSCK9 (HR: 0.57, 95% CI: 0.38–0.85, *p* = 0.006), but not for those without the elevation of either parameter. Interactions remained significant between LL-37 and lp(a) (*p* _interaction_ = 0.036), but were not statistically significant between LL-37 and PCSK9 (*p* _interaction_ = 0.094). The prognostic impacts of LL-37 were insignificant and not altered despite varying levels of LDL-C or triglyceride. Interestingly, high LL-37 was associated with lower risk for patients with reduced HDL-C (HR: 0.57, 95% CI: 0.35–0.91, *p* = 0.018).

For all-cause death (Figure 6), high LL-37 was associated with lower mortality only if lp(a) (HR: 0.36, 95% CI: 0.14–0.95, *p* = 0.040) or PCSK9 (HR: 0.53, 95% CI: 0.29–0.98, *p* = 0.043) levels were elevated. There was a trend for significant interactions between LL-37 and lp(a) (*p* _interaction_ = 0.051), but not for PCSK9 (*p* _interaction_ = 0.261).

Similar disparities in HR values were seen for outcomes of cardiac death, recurrent MI, and ischemic stroke (Figure 7, Figure 8 and Figure 9), as patients with high LL-37 and elevated lp(a) or PCSK9 acquired a numerically lower risk, although without statistical significance (all *p* > 0.05) or substantial interactions (all *p* _interaction_ > 0.05). Meanwhile, the prognostic impacts of LL-37 were generally insignificant and not altered when stratified by levels of LDL-C or triglyceride. Interestingly, high LL-37 was associated with a substantially lower risk of ischemic stroke (HR: 0.40, 95% CI: 0.17–0.92, *p* = 0.030) in patients with reduced HDL-C.

## 4. Discussion

Previously, our team had reported in a small prospective cohort of STEMI patients that high levels of LL-37 were associated with lower ischemic risk. In the current study, we further discovered that such a risk reduction associated with LL-37 was significantly affected by circulating levels of lp(a) and PCSK9. These findings suggest that LL-37 could be a potential new therapeutic option to tackle residual ischemic risk due to elevations of lp(a) and PCSK9.

### 4.1. Interactions between LL-37 and lp(a)

Despite controversies regarding its synthesis and metabolism, the pathogenic role and prognostic impact of lp(a) have been established in the field of atherosclerosis and coronary heart disease after decades of research [4]. Evidence from large-scale cohort studies demonstrates the increased risk of atherosclerotic cardiovascular disease for patients with elevated lp(a), which is mainly attributed to its pro-inflammatory and prothrombotic effects by contents of oxidized phospholipids (OxPL) and apolipoprotein(a) [15]. However, medications to tackle cardiovascular risk caused by lp(a) elevation are quite limited. So far, lipoprotein apheresis is proven to effectively reduce circulating lp(a) levels by 53% to 73% from baseline, followed by 23% with PCSK9 inhibitors, while traditional statin medications even paradoxically drive up lp(a) concentrations up to 30% [4]. Dietary, lifestyle, and medical interventions are generally ineffective for reducing lp(a), as 90% of its plasma levels are genetically determined and remain stable throughout a lifetime, for which nucleic acid-based therapies have shown promising results. Moreover, lp(a)-related ischemic risk could be further enhanced by systematic inflammation, which is commonly presented in MI patients [16]. Therefore, the continuous search for potential agents to tackle the residual lipid risk caused by lp(a) is of great clinical importance, especially for the secondary prevention of MI patients. Previous research shows that LL-37 downregulates the expression of CD36 receptors in adipocytes and hepatocytes, leading to a reduction in lipid accumulation and hepatic steatosis [12]. Other studies also report correlations between LL-37 and various lipid contents [9,17]. These findings suggest possible interactions between LL-37 and lipid metabolism. However, the clinical impacts of LL-37 in AMI patients with dyslipidemia are still unknown. The current study shows that high levels of LL-37 were potentially beneficial for patients with elevated lp(a). For patients with lp(a) above 300 mg/L, high LL-37 was associated with a 51% lower risk of MACE, which was not observed in patients with lp(a) within a normal range, and significant interactions affirmed that varying prognostic impacts of LL-37 were associated with baseline lp(a) levels. Such disparities in risk reduction were sustained in secondary outcome analysis, although with attenuated statistical significance.

Interpretations for the above findings could be challenging, but the unique synthesis and metabolism of lp(a) could be a major contributor. Since dyslipidemia could be effectively controlled with intensive secondary preventions, risk reduction associated with LL-37 might not be detected when stratifying patients with LDL-C, HDL-C, or triglyceride, as these lipid contents could be restored to safer ranges with adequate lipid-lowering medications [5]. However, lifelong exposure to high lp(a) determined by genes is hard to resolve with currently available medications, while lp(a) particles acquire a more atherogenic nature than LDL-C particles [4,5,15]. OxPL is proven to play a major role in arterial inflammation induced by lp(a), as it could predispose in the vessel wall, promote monocyte infiltration, and increase the secretion of pro-inflammatory cytokines (e.g., interleukin (IL)-8, monocyte chemoattractant protein-1), causing endothelial dysfunction, lipid accumulation, and regional inflammation within coronary plaques [4,15]. Meanwhile, over 80% of circulating LL-37 bind with a wide range of lipoproteins, such as LDL or very low-density lipoprotein [10,18]. However, the current study has not detected significant correlations between LL-37 and lp(a). Therefore, the observed risk reduction associated with LL-37 was more likely achieved through its interactions with lp(a) to reduce its downstream atherogenic effects on vessel walls, rather than reducing lp(a) synthesis or release into the circulation. Notably, the activation and expression of several potential receptors for lp(a) (e.g., toll-like receptors (TLR), CD36) could be effectively inhibited by LL-37, which limits the downstream activation of inflammatory signals (e.g., Myd88, NF-κB, IL-1β) and the subsequent risk of atherothrombosis due to chronic inflammatory response [12,19,20]. Previous research has also shown that LL-37 could prevent the binding between ligands and TLR 1/2/4/6, which are essential targets of oxPL to induce the activation of pro-inflammatory intracellular signal pathways [20,21,22]. Taken together, ischemic risk due to inflammatory stimulus of active oxPL might not be effectively tackled using conventional lipid-lowering or antiplatelet medications, for which LL-37 might offer extra atheroprotection through its suppression of regional and systemic inflammation due to the long-term exposure of elevated lp(a). More research from bench to bedside is needed to validate biological interactions between LL-37 with various lipoproteins and their receptors.

### 4.2. Interactions between LL-37 and PCSK9

As an important regulator of cholesterol, PCSK9 drives up circulating levels of LDL-C by downregulating LDL receptors (LDLRs) [23]. The newly developed monoclonal antibody for PCSK9 allows patients with dyslipidemia to attain LDL-C levels as low as 0.78 mmol/L [24]. However, PCSK9 inhibitors are not routinely used in current clinical practice, as they bring about concerns regarding adverse effects due to reaching extremely low LDL-C, the inconvenience of injections, and increased medical costs [2,24]. On the other hand, statin remains the first-line and mainstream lipid-lowering agent for patients with atherosclerosis, but this classical medication inevitably drives up levels of PCSK9 within the circulation [3,23]. Furthermore, elevations of PCSK9 also contribute to increased levels of lp(a), which could further increase the risk of ischemic events [4,25]. Therefore, it is of great clinical interest for developing more effective agents to tackle ischemic risk caused by high levels of circulating PCSK9. The current study detected a significant risk reduction associated with a high level of LL-37 for AMI patients with PCSK9 levels above the median, where the relative risk of MACE was substantially reduced by 43%, which was not seen in patients with lower levels of PCSK9.

Mechanisms for improved outcomes associated with high LL-37 among patients with elevated PCSK9 could be multiple, but relevant reports are quite scarce. The current study detected a significant positive correlation between levels of PCSK9 and LL-37, indicating rigorous interactions between these two molecules. Meanwhile, it should be emphasized that circulating levels of PCSK9 are largely determined by upstream gene expressions, and life-long exposure to elevated PCSK9 could not be effectively attenuated with lifestyle changes and conventional lipid-lowering medications [23,26]. Therefore, it seemed less likely that LL-37 could regulate the expression of PCSK9 and further reduce LDL-C to achieve lower ischemic risk.

Based on current knowledge, LL-37 might interfere with PCSK9 through the following pathways. Firstly, LL-37 could possibly abrogate the binding between PCSK9 and LDLR, therefore restoring cholesterol receptors on the cell surface and inhibiting ongoing atherosclerosis. The recycling of LDLR induced by PCSK9 is mainly achieved through the binding between the epidermal growth factor-like repeat-A (EGF-A) domain on LDLR and the catabolic domain on PCSK9, for which PCSK9 inhibitors have been developed to bind with EGF-A site on PCSK9, and therefore silence its functions, prevent degradations of LDLR, and therefore reduce LDL-C levels [27]. In recent years, several peptide inhibitors for PCSK9 have been found or developed, including both analogs of EGF-A and non-EGF-A-based peptides identified from randomized peptide sequences, which are proven to effectively bind with the EGF-A site on PCSK9 and inhibit the PCSK9:LDLR interface with high potency [27,28]. More recently, PCSK9 has been found to be actively involved in host defense in infectious diseases (e.g., sepsis, hepatitis), so there could be possible interactions between PCSK9 and various immunomodulators, including antimicrobial peptide LL-37 [29,30] Future studies are warranted to investigate whether LL-37 could bind with PCSK9 to restore the function of LDLR and its impacts on long-term LDL-C levels, which would provide essential evidence about whether LL-37 could be a potential antagonist for PCSK9.

Another possible pathway might involve LL-37’s inhibition of inflammatory response induced by elevated PCSK9. Accumulating evidence from basic research shows that PCSK9 promotes vascular inflammation through upregulating classical pro-inflammatory pathways involved in atherosclerosis, such as TLR4/MyD88/NF-κB and NLRP3 inflammasomes [31,32]. Evidence from clinical research also demonstrates that elevated PCSK9 is associated with systematic inflammation and platelet activation, while high levels of PCSK9 predict a greater risk of recurrent ischemia, especially for those with diabetes and ongoing inflammation [33,34,35]. Therefore, PCSK9 not only leads to greater lipid risk but could also drive up ischemic events through inflammatory and thrombotic pathways. On the other hand, LL-37 is proven to inhibit the response of pro-inflammatory macrophages, the activation of TLRs, and downstream pro-inflammatory signals (e.g., IL-1β, IL-6, TNF-α) [19,20,36,37]. In vivo studies have shown the potential of LL-37 to suppress atherosclerosis and in-stent restenosis by reducing excessive inflammatory response and promoting tissue repair. Mihailovic et al. report that low-dose cathelicidin-related antimicrobial peptide (CRAMP), which shares 46% amino acid sequence identity with human LL-37, significantly reduces aortic plaque burden in ApoE-/- mice on a high-fat diet, which involves CD8+ T-cell degranulation and reduction in CD11b+/CD11c+ conventional dendritic cells [38]. Soehnlein et al. demonstrate that LL-37 could promote re-endothelization and limit neointima formation after stent implantation in ApoE-/- mice, which is mediated by the enhanced recruitment and functions of early outgrowth cells [39]. The pathobiology of AMI is mainly due to atherothrombosis induced by the rupture and erosion of unstable coronary plaques, for which revascularization and vascular healing play central roles in preventing recurrent ischemia [40]. In the current cohort, the majority of patients were diagnosed with STEMI and successfully treated with primary PCI, and ischemic risk reduction from LL-37 could therefore be mediated by its suppression of arterial inflammation, the inhibition of neo-atherosclerosis, promotion for vessel recovery, and the prevention of restenosis.

In sum, high levels of LL-37 proved to be associated with a lower risk of MACE in AMI patients with elevated lp(a) or PCSK9. Although the mechanisms behind this were still unclear, available evidence suggests that the atheroprotection from LL-37 might be attributed to its direct interference with PCSK9 or lp(a) and its inhibition of downstream inflammatory responses.

### 4.3. Limitations

The major limitations of the current study are as follows. Firstly, this study was completed by a single center. Although the sample size was large enough, the extrapolation of conclusions might be limited due to variations in ethnicity, region, and clinical practice. Future studies from other populations and datasets are needed to validate the results from the current report. Secondly, this study acquired an observational design. Despite multiple adjustments for potential confounding, the causality between LL-37, lipid profiles, and clinical outcomes could not be fully defined, as undetected bias could still affect the results. The attenuated significance for several outcomes suggested a possibility of finding by chance. Finally, the dynamic changes of LL-37 and lipid profiles were not available in the current study. More evidence is needed regarding how LL-37 could affect changes in lp(a) and PCSK9 during the follow-up.

## 5. Conclusions

For patients with AMI, the risk reduction associated with LL-37 varied according to levels of lp(a) and PCSK9. High levels of LL-37 were associated with a lower risk of MACE in AMI patients with elevated lp(a) or PCSK9.

## Figures and Tables

**Figure 1 biomolecules-12-01482-f001:**
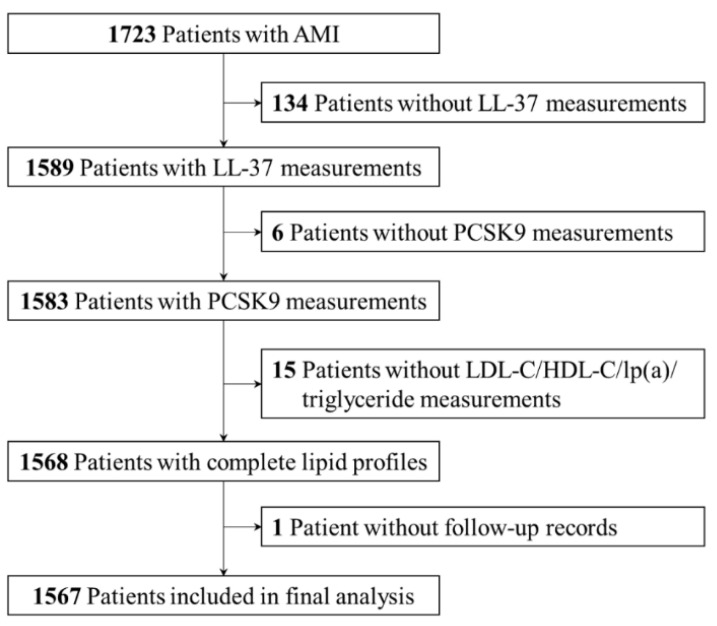
Study flow chart. AMI = acute myocardial infarction; HDL-C = high-density lipoprotein cholesterol; LDL-C = low-density lipoprotein cholesterol; lp(a) = lipoprotein(a); PCSK9 = proprotein convertase subtilisin/kexin type 9.

**Figure 2 biomolecules-12-01482-f002:**
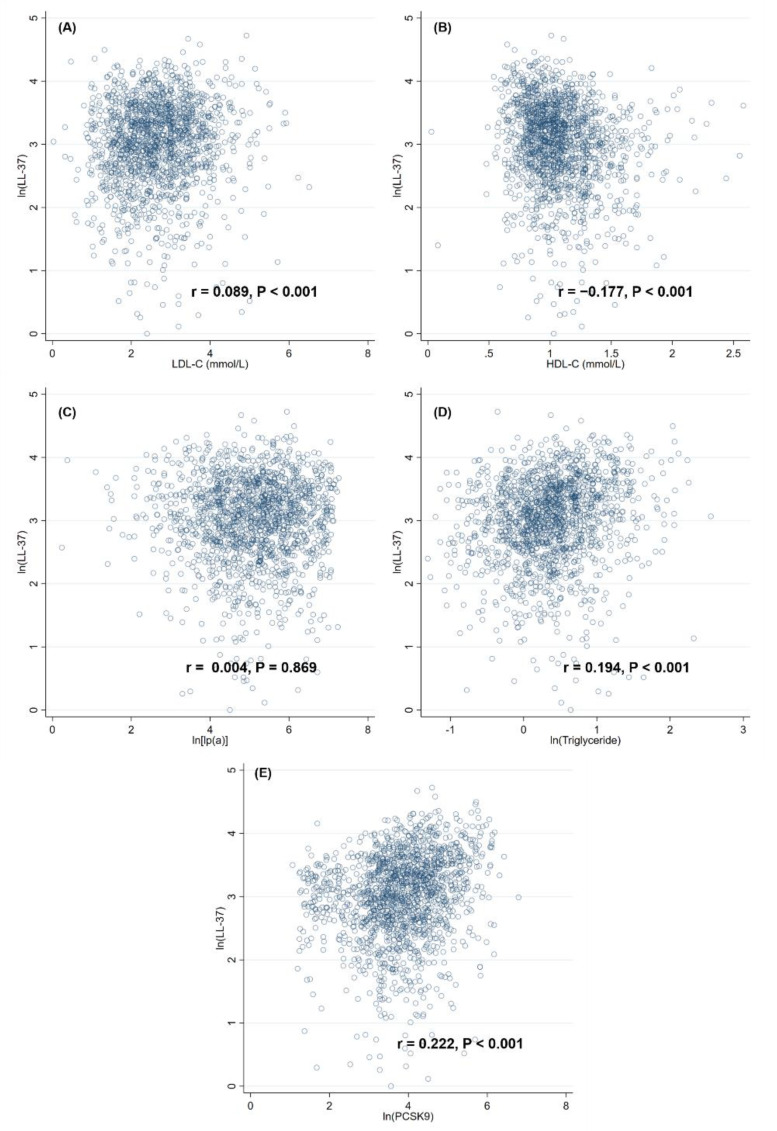
Correlations between levels of LL-37 and various lipid contents, including LDL-C (**A**), HDL-C (**B**), lp(a) (**C**), triglyceride (**D**) and PCSK9 (**E**). HDL-C = high-density lipoprotein cholesterol; LDL-C = low-density lipoprotein cholesterol; lp(a) = lipoprotein(a); PCSK9 = proprotein convertase subtilisin/kexin type 9.

**Figure 3 biomolecules-12-01482-f003:**
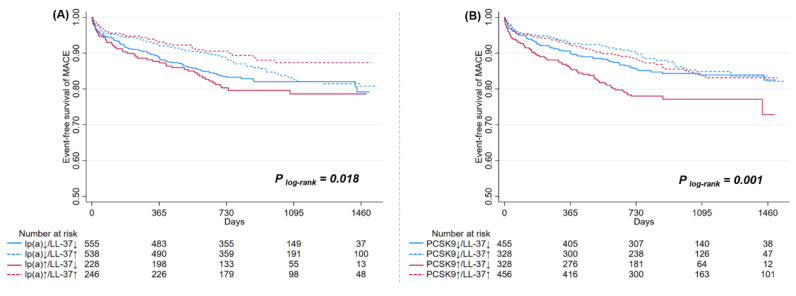
Associations between LL-37 and MACEs according to levels of lp(a) (**A**) and PCSK9 (**B**). MACEs = major adverse cardiovascular events; lp(a) = lipoprotein(a); PCSK9 = proprotein convertase subtilisin/kexin type 9; ↓ = low; ↑ = high.

**Figure 4 biomolecules-12-01482-f004:**
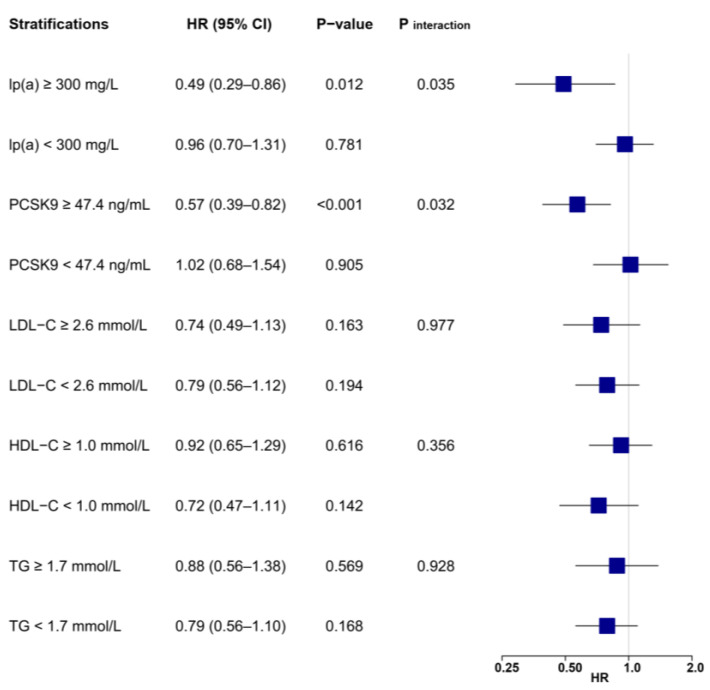
Associations between LL-37 and the risk of MACE according to lipid profile values. Multivariable Cox regression was adjusted for variables with *p* < 0.1 in univariable analysis (see Appendix A), including age, gender, body mass index, hypertension, previous myocardial infarction, heart rate, ejection fraction, leucocytes, hemoglobin, creatinine, glucose, hemoglobin A1c, high-sensitivity C-reactive protein, cardiac troponin I, N-terminal prohormone of brain natriuretic peptide, LDL-C, culprit lesion, multivessel disease, successful revascularization, P2Y12 inhibitor, and statin. HDL-C = high-density lipoprotein cholesterol; LDL-C = low-density lipoprotein cholesterol; lp(a) = lipoprotein(a); MACE = major adverse cardiovascular event; PCSK9 = proprotein convertase subtilisin/kexin type 9; TG = triglyceride.

**Figure 5 biomolecules-12-01482-f005:**
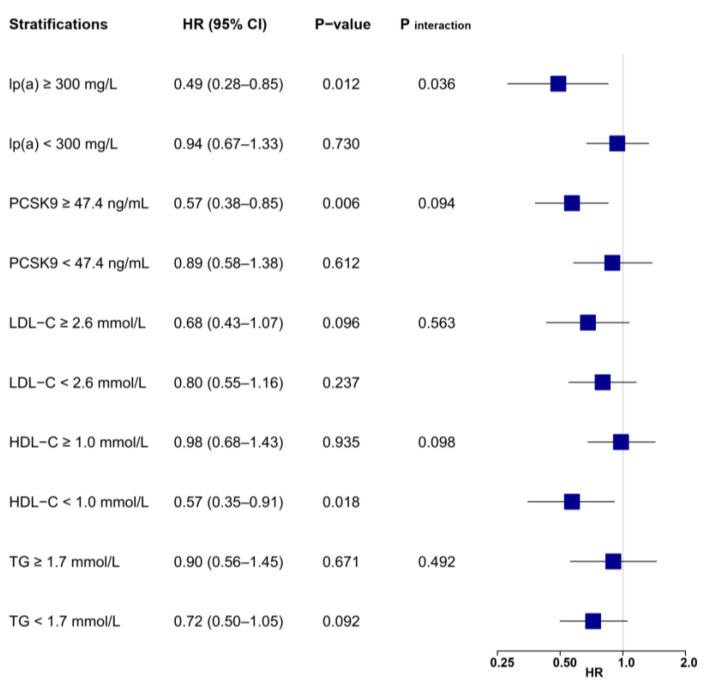
Associations between LL-37 and the risk of secondary composite endpoint (cardiac death, recurrent MI, and ischemic stroke) according to lipid profile values. Multivariable Cox regression was adjusted for variables with *p* < 0.1 in univariable analysis (see Appendix A), including age, body mass index, hypertension, previous MI, heart rate, ejection fraction, hemoglobin, creatinine, glucose, hemoglobin A1c, high-sensitivity C-reactive protein, cardiac troponin I, N-terminal prohormone of brain natriuretic peptide, LDL-C, multivessel diseases, successful revascularization, and P2Y12 inhibitor. HDL-C = high-density lipoprotein cholesterol; LDL-C = low-density lipoprotein cholesterol; lp(a) = lipoprotein(a); MI = myocardial infarction; PCSK9 = proprotein convertase subtilisin/kexin type 9; TG = triglyceride.

**Figure 6 biomolecules-12-01482-f006:**
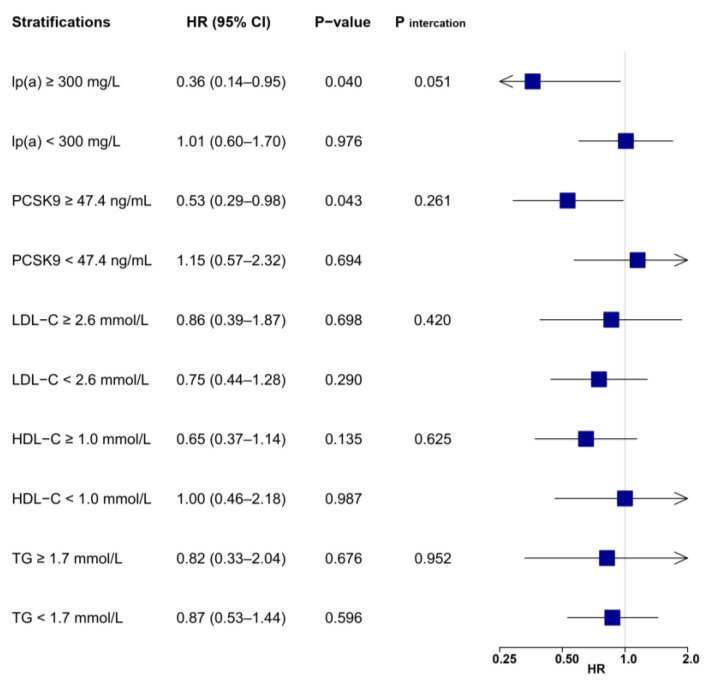
Associations between LL-37 and the risk of all-cause death according to lipid profile values. Multivariable Cox regression was adjusted for variables with *p* < 0.1 in univariable analysis (see Appendix A), including age, gender, body mass index, hypertension, diabetes, previous myocardial infarction, heart rate, systolic BP, diastolic BP, ejection fraction, leucocytes, hemoglobin, creatinine, glucose, hemoglobin A1c, high-sensitivity C-reactive protein, cardiac troponin I, N-terminal prohormone of brain natriuretic peptide, LDL-C, TG, culprit lesion, multivessel diseases, successful revascularization, aspirin, P2Y12 inhibitor, and statin. BP = blood pressure; HDL-C = high-density lipoprotein cholesterol; LDL-C = low-density lipoprotein cholesterol; lp(a) = lipoprotein(a); PCSK9 = proprotein convertase subtilisin/kexin type 9; TG = triglyceride.

**Figure 7 biomolecules-12-01482-f007:**
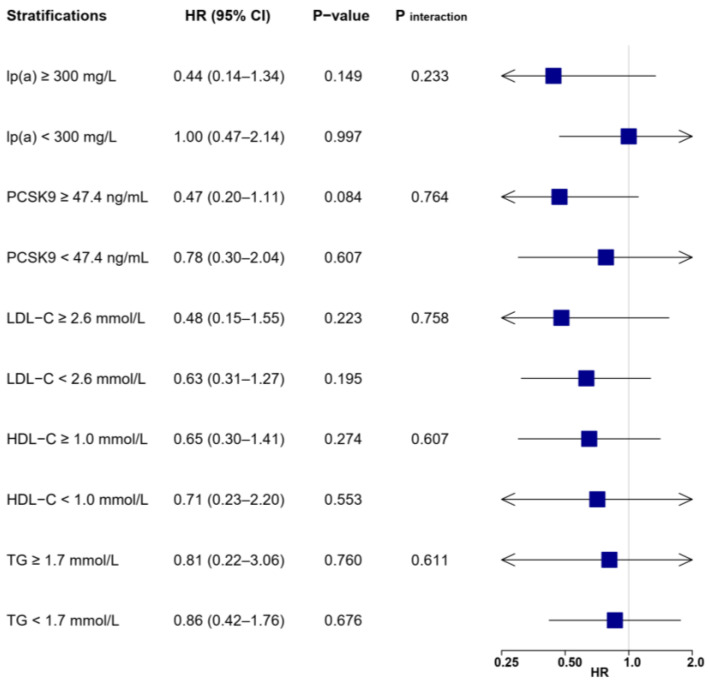
Associations between LL-37 and the risk of cardiac death according to lipid profile values. Multivariable Cox regression was adjusted for variables with *p* < 0.1 in univariable analysis (see Appendix A), including age, gender, body mass index, hypertension, diabetes, previous myocardial infarction, heart rate, systolic BP, diastolic BP, ejection fraction, leucocytes, hemoglobin, creatinine, glucose, hemoglobin A1c, high-sensitivity C-reactive protein, cardiac troponin I, N-terminal prohormone of brain natriuretic peptide, LDL-C, lp(a), culprit lesion, multivessel diseases, successful revascularization, P2Y12 inhibitor, and statin. BP = blood pressure; HDL-C = high-density lipoprotein cholesterol; LDL-C = low-density lipoprotein cholesterol; lp(a) = lipoprotein(a); PCSK9 = proprotein convertase subtilisin/kexin type 9; TG = triglyceride.

**Figure 8 biomolecules-12-01482-f008:**
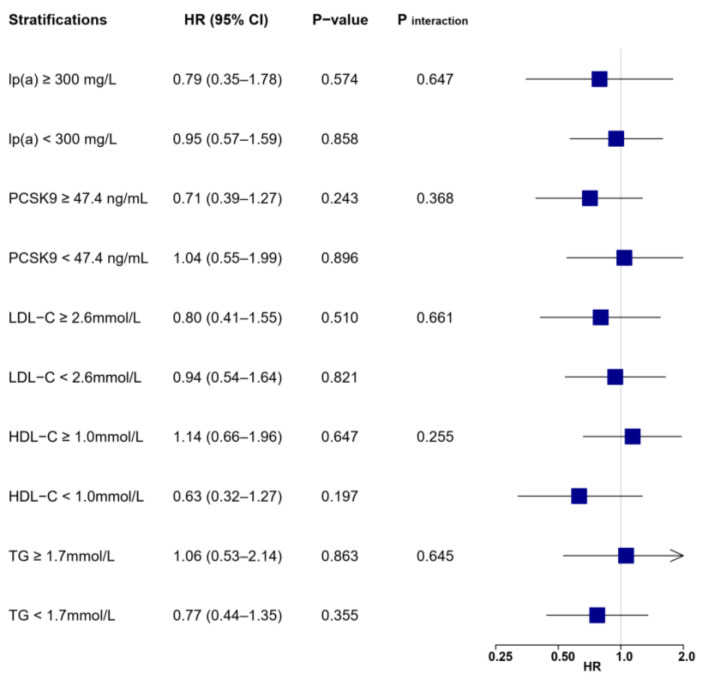
Associations between LL-37 and the risk of recurrent MI according to lipid profile values. Multivariable Cox regression was adjusted for variables with *p* < 0.1 in univariable analysis (see Appendix A), including age, previous MI, systolic blood pressure, creatinine, N-terminal prohormone of brain natriuretic peptide, LDL-C, and successful revascularization. HDL-C = high-density lipoprotein cholesterol; LDL-C = low-density lipoprotein cholesterol; lp(a) = lipoprotein(a); MI = myocardial infarction; PCSK9 = proprotein convertase subtilisin/kexin type 9; TG = triglyceride.

**Figure 9 biomolecules-12-01482-f009:**
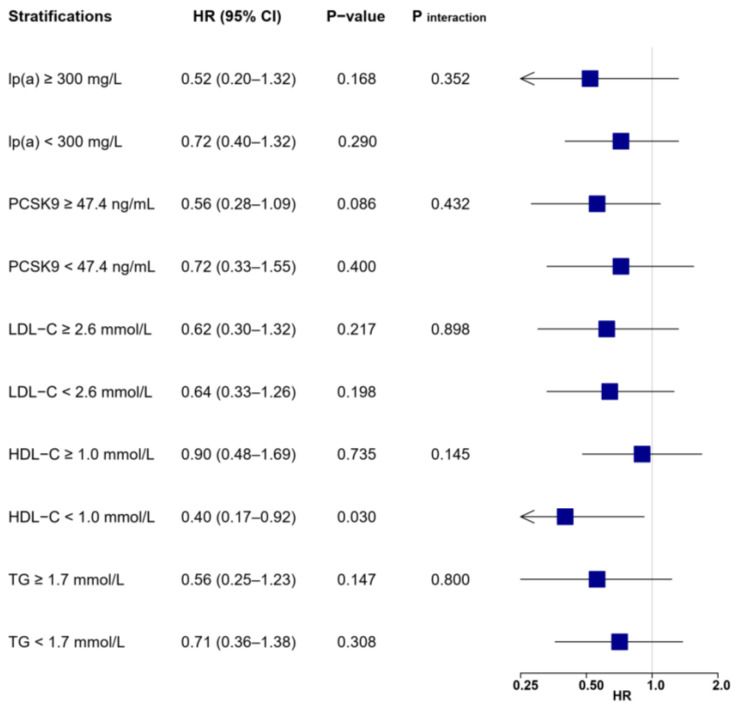
Associations between LL-37 and the risk of ischemic stroke according to lipid profile values. Multivariable Cox regression was adjusted for variables with *p* < 0.1 in univariable analysis (see Appendix A), including body mass index, hypertension, diabetes, previous myocardial infarction, systolic BP, diastolic BP, ejection fraction, glucose, hemoglobin A1c, N-terminal prohormone of brain natriuretic peptide, LDL-C, and successful revascularization. BP = blood pressure; HDL-C = high-density lipoprotein cholesterol; LDL-C = low-density lipoprotein cholesterol; lp(a) = lipoprotein(a); PCSK9 = proprotein convertase subtilisin/kexin type 9; TG = triglyceride.

**Table 1 biomolecules-12-01482-t001:** Baseline characteristics stratified by median level of LL-37.

Variables	All (N = 1567)	LL-37 <21.86 ng/mL (N = 783)	LL-37 ≥21.86 ng/mL (N = 784)	*p*-Value
Age, years	60.7 ± 12.3	62.3 ± 12.4	59.1 ± 12.1	<0.001
Male sex, n (%)	1263 (80.6)	602 (76.9)	661 (84.3)	<0.001
BMI, kg/m^2^	25.9 ± 3.8	25.7 ± 4.0	26.1 ± 3.5	0.022
Hypertension, n (%)	1021 (65.2)	511 (65.3)	510 (65.1)	0.93
Diabetes, n (%)	541 (34.5)	282 (36.0)	259 (33.0)	0.21
Smoking, n (%)	1118 (71.3)	538 (68.7)	580 (74.0)	0.021
Previous MI, n (%)	272 (17.4)	144 (18.4)	128 (16.3)	0.28
STEMI, n (%)	1389 (88.6)	685 (87.5)	704 (89.8)	0.149
Hemodynamics				
Heart rate, bpm	76.1 ± 15.0	75.7 ± 14.4	76.5 ± 15.5	0.31
SBP, mmHg	125.7 ± 19.7	125.7 ± 19.8	125.6 ± 19.7	0.86
DBP, mmHg	78.5 ± 13.3	77.7 ± 12.9	79.4 ± 13.8	0.011
EF, %	53.7 ± 7.6	53.6 ± 7.7	53.7 ± 7.6	0.63
Lab test				
Hemoglobin, g/L	139.9 ± 19.7	138.0 ± 19.8	141.8 ± 19.5	<0.001
Leucocytes, ×10^9^/L	9.5 ± 3.3	9.2 ± 3.3	9.9 ± 3.2	<0.001
Platelets, ×10^9^/L	229.8 ± 79.1	224.9 ± 79.5	234.7 ± 78.4	0.014
Creatinine, μmol/L	89.3 ± 36.9	89.9 ± 41.1	88.7 ± 32.3	0.52
Glucose, mmol/L	8.4 ± 3.8	8.4 ± 3.7	8.3 ± 3.8	0.42
HbA1c, %	6.7 ± 1.6	6.7 ± 1.6	6.7 ± 1.5	0.65
hsCRP, mg/L	5.62 (1.97–10.94)	5.09 (1.82–10.75)	6.09 (2.13–11.08)	0.040
cTnI, ng/mL	0.914 (0.108–5.000)	1.030 (0.113–5.760)	0.810 (0.101–4.437)	0.073
NT-proBNP, pg/mL	281.5 (71.6–978.0)	337.4 (89.5–1182.1)	233.7 (55.4–754.75)	<0.001
LDL-C, mmol/L	2.7 ± 0.9	2.6 ± 0.9	2.8 ± 0.9	<0.001
HDL-C, mmol/L	1.1 ± 0.3	1.1 ± 0.4	1.0 ± 0.3	<0.001
Lp(a), mg/L	169.0 (74.6–355.0)	157.0 (76.0–343.0)	179.1 (71.8–375.7)	0.37
Triglyceride, mmol/L	1.45 (1.02–2.05)	1.30 (0.94–1.79)	1.58 (1.10–2.30)	<0.001
LL-37, ng/mL	21.86 (13.89–32.67)	13.89 (9.17–17.86)	32.66 (26.87–41.53)	<0.001
PCSK9, ng/mL	47.40 (26.11–89.97)	38.85 (22.51–72.40)	59.76 (30.57–107.44)	<0.001
Angiography and treatments				
Culprit lesion, n (%)				
LM	13 (0.8)	4 (0.5)	9 (1.1)	0.53
LAD	718 (45.8)	359 (45.8)	359 (45.8)	
LCX	227 (14.5)	121 (15.5)	106 (13.5)	
RCA	596 (38.0)	293 (37.4)	303 (38.6)	
Bypass graft	13 (0.8)	6 (0.8)	7 (0.9)	
Multivessel disease, n (%)				
One-vessel disease	393 (25.1)	195 (24.9)	198 (25.3)	0.93
Two-vessel disease	457 (29.2)	226 (28.9)	231 (29.5)	
Three-vessel disease	717 (45.8)	362 (46.2)	355 (45.3)	
Successful revascularizations ^1^, n (%)	1389 (88.6)	692 (88.4)	697 (88.9)	0.74
Medications ^2^				
Aspirin, n (%)	1506 (96.1)	751 (95.9)	755 (96.3)	0.69
P2Y12 inhibitor, n (%)	1544 (98.5)	774 (98.9)	770 (98.2)	0.29
Statin, n (%)	1496 (95.5)	745 (95.1)	751 (95.8)	0.54

BMI = body mass index; cTnI = cardiac troponin I; DBP = diastolic blood pressure; EF = ejection fraction; HbA1c = hemoglobin A1c; HDL-C = high-density lipoprotein cholesterol; hsCRP = high sensitivity C-reactive protein; LAD = left anterior descending artery; LCX = left circumflex; LDL-C = low-density lipoprotein cholesterol; LM = left main; lp(a) = lipoprotein(a); NT-proBNP = N-terminal prohormone of brain natriuretic peptide; PCSK9 = proprotein convertase subtilisin/kexin type 9; RCA = right coronary artery; SBP = systolic blood pressure; STEMI = ST-segment elevation myocardial infarction. ^1^: Revascularizations included percutaneous coronary intervention and coronary bypass grafting. ^2^: Medications typically referred to drugs prescribed at discharge, or otherwise, drugs being used during hospitalization if patients failed to survive the hospitalization.

## Data Availability

The data used to support the findings of this study are available from the corresponding authors upon request. The institution (Fuwai Hospital) requires all requests for accessing any data of patients to be applied for and processed in a case-by-case manner.

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
