# Peer review of "Prognostic Impacts of LL-37 in Relation to Lipid Profiles of Patients with Myocardial Infarction: A Prospective Cohort Study"

_biomolecules, 2022, doi:10.3390/biom12101482_

Round 1

Reviewer 1 Report

Dear Authors and Editor,

This is a well conducted and described studies addressing the association and prognostic value of LL37 in patients with AMI. The authors found a nominally significant, but consistent, positive interactions of LL37 with Lp(a) and PCSK9 on their association with MACE.

I have the following major comments:

 1. Given the observational design of the study, avoid using terms such as “protective effect” please use the terms “association” or “prognostic value” instead.

2. Are the “LL37 by Lp(a)” and “LL37 by PCSK9” independent from each others? This could be tested in two ways, a formal one that is including both interactions terms in the models to see if they both are significant, and second one would be to create 4 groups of patients (using Lp(a) and PCSK9 categories to see if the LL37 association with MACE is similar or not in these groups (testing LL37 by PCSK9-LPa group interactions). Please perform these analyses and include them in the manuscript.

3. Conclusions should be interpreted with caution, indeed, given the multiple hypothesis being tested, the P values for interaction LL37 by Lp(a) or LL37 by PCSK9 would not survive adjustment for inflation of alpha error (e.g. Bonferroni adjustment). Moreover the lack of replication in other dataset warrants caution. Please discuss these limitations appropriately in the discussion.

4. Among the limitations, it should be also added that the lack of measures of LL37 changes over time is another point not allowing to infer causality from these associations between LL37 and MACE and death.

5. I suggest the authors consider that the same interaction can be seen also as the other way around, therefore PCSK9 and lp(a) association with MACE are also influenced by LL-37 levels. It should be helpful to show the interaction also from this point of view. Overall, this suggests that LL37 levels are relevant only when Lp(a) or PCSK9 are elevated, therefore suggesting synergistic actions of these factors on MACE. Please comments.

Reviewer 2 Report

The authors have chosen a definition of MACE including all-cause death. It should be better to include, rather, cardiovascular death. 

Presentation of Statistical methods does not include the number of participants or of events necessary

The authors have chosen a presentation of their data stratified accordin to LL-37 median, wich might be fine. Many confounders are found. Most importantly, the authors do not fully take these confounders into account.

Figures: survival curves stratified according to LLC and lipid levels should also be adjusted for most confounders (age, BMI, gender, etc).

Not clear for me if Table 2 shows outcome associated with LL-37 or lipid levels.

Discussion: This is an observationnal study that  does not support any causality link between LL-37 levels and lipid impact on occurrence of MACE.  The authors must be much more careful in their conclusions

Reviewer 3 Report

The presented work  aimed to investigate the connection between various lipid contents and  protective effects of LL-37 in patients with AMI. The data are clean and adds some new important data on the possible relation of LL-37 protective effect  to lipoproteins. Only minor changes are suggested. Due to the nature of the variables distribution the simple correlations of the LL-37 and lipid  related parameters, I propose to use the Spearman  instead of  the Pearson correlation. Additionally, it would be worth paying attention to the very low correlation coefficient, despite the significance of the test used.

Addition of a short description of the applied statistical methods directly next to the tables and figures would make it much easier to track the results and correctness of the conclusions. Particularly unclear is Table 2.

Reviewer 4 Report

Concerning this manuscript, some doubts arose that need to be clarified and some modifications must be made.

- In the sentences “…hypercholesterolemia, is one of the most important risk factors for patients with coronary artery disease (CAD)” and “… are at the highest risk levels among CAD patients among CAD patients.” What risk do the authors refer to? Did the authors mean “…hypercholesterolemia, is one of the most important risk factors for coronary artery disease (CAD) development” “Patients with myocardial infarction (MI) have a higher risk of developing CAD”?

Other unclear sentences or fragments of a sentence: “…indicating high prevalence of residual risk of cholesterol”, “LL-37 is able to reduce ... lipid accumulation of adipocytes and hepatocytes”, “higher levels of LL-37 was associated lower risk of MACE”, among others.

“Lipid risk” refers to atherogenic lipid levels (or cardiovascular risk profile)?

An English revision should be considered.

- At introduction, PCI is used without previous definition (presented posteriorly at Materials and Methods).

-It was stated “…high level of LL-37 is a protective factor…”, the aim was to state “high level of LL-37 is a predictor of lower risk of ischemic cardiovascular events”? According to the reference cited (doi: 10.5551/jat.63221) LL-37 was reported to be a protective factor or a predictor biomarker? A clarification is needed. If LL-37 has protective effect, which mechanism was pointed to explain how high LL-37 induces this protective role/effect?

- At introduction, it is stated “LL-37 is able to reduce expression of CD36”, but the importance of this reduction is not referred. Hoang-Yen Tran et al. (doi: 10.1038/ijo.2016.90) reported that cathelicidin (LL-37 and mCRAMP) inhibits the CD36 fat receptor, which probably contributes to suppress lipid accumulation in adipocytes and hepatocytes, leading to a reduction of fat mass and hepatic steatosis.

- At Material and Methods, the methods used for the evaluation of lipid profile were not presented.

- At Material and Methods, an explanation of the cut-of value used for PCSK9 levels (high vs low, ≥ 47.4 ng/mL vs < 47.4 ng/mL) should be provided.

- In Results, at table 1, are presented determinations that were not referred at Material and Methods (as blood cell count, creatinine, glucose, HbA1c, hs-CRP, cTNI, NT-proBNP), neither the methods used for their evaluation. Evaluation of the hemodynamics parameters was not referred either at Material and Methods. Leukocyte differential count is not available?

- It was stated “According to univariable Cox regression, higher levels of LL-37 was associated lower risk of MACE (hazard ratio [HR]: 0.71, 95 % confidence interval [CI]: 0.55- 0.91, P = 0.007), but its statistical significance was attenuated (HR: 0.68, 95 % CI: 0.41-1.14, P = 0.141)”, according to P-value, statistical significance was lost not attenuated.

- It was stated “Similar results were seen for outcomes of cardiac death, recurrent MI and ischemic stroke, as patients with high LL-37 and elevated lp(a) or PCSK9 acquired lower risk, although with attenuated statistical significance”; the P-value must be presented.

- According to the authors, some variables did not present a normal distribution (as triglycerides, lp(a), LL-37, PCSK9); therefore, why the authors did not used the Spearman correlation test? Besides, the use of the Pearson correlation test, which results are presented at figure 2, was not mentioned in Statistical Analysis, at Results.

- At discussion, it was stated “…lp(a), … contents of oxidized phospholipids (OxPL) and apolipoprotein A”, but lp(a) contains apolipoprotein (a) (not to be confused with apo A1).

- Data obtained do not allow to state “In sum, the more significant risk reduction from high LL-37 in AMI patients with high lp(a) suggested that LL-37 acquire the potential to tackle residual lipid risk of lp(a) that might not be adequately resolved with traditional statin treatment” (the sentence is unclear, and the idea may have been misinterpreted).

- At discussion was referred “we further discovered that atheroprotective effect associated to increased LL-37”; explain how data obtained allows to attribute an atheroprotective protective effect for high LL-37 and by which mechanism(s) LL-37 exerts a protective role. A detailed explanation of the reason why high levels of LL-37 are potentially beneficial for patients with AMI and elevated lp(a) or PCSK9 levels but not for those with low levels of these two markers is needed. Is LL-37 a biomarker (a predictor marker) in patients with AMI and elevated lp(a) or PCSK9 of MACE, or LL-37 has an interventional role (a protective effect) reducing the risk of MACE in individuals with an enhancement in its levels and with AMI and increase of lp(a) or PSCK9? Hypothesizing that the enhancement of LL-37 is a response to the inflammation promoted by lp(a), mainly observed when lp(a) is increased, which is the mechanism, the pathway, involved that results in an apparent protective role? Patients with AMI, high LL-37 and high lp(a) or high PCSK-9 present higher values of hs-CRP and leukocyte count than subjects with AMI, high LL-37 and low lp(a)/PSCK9? Discussion must respond to these questions.

Beside, prudence is needed at discussion/conclusions, since statistical significance was lost after multiple adjustments, as stated by the authors, “According to univariable Cox regression, higher levels of LL-37 was associated lower risk of MACE (hazard ratio [HR]: 0.71, 95 % confidence interval [CI]: 0.55- 0.91, P = 0.007), but its statistical significance was attenuated (HR: 0.68, 95 % CI: 0.41-1.14, P = 0.141) after multiple adjustments for other baseline variables with P < 0.1 in univariable analysis” and “Similar results were seen for outcomes of cardiac death, recurrent MI and ischemic stroke, as patients with high LL-37 and elevated lp(a) or PCSK9 acquired lower risk, although with attenuated statistical significance.”

- Abstract should be altered in accordance with the modifications made in the manuscript.

Round 2

Reviewer 1 Report

Dear Authors ,

thanks for your reply to previous comments. The manuscript is overall improved.

Please modify the following sentence: "In sum, high levels of LL-37 proved to be clinically beneficial for AMI patients with 414 elevated lp(a) or PCSK9."

This study shows associations, not causality that is yet to be proven. 

Author Response

We sincerely thank for the endorsement for our response and careful consideration by the reviewer. We have accordingly modified the relevant sentence:

(Page 15, line 427-428) In sum, high levels of LL-37 proved to be associated with lower risk of MACE in AMI patients with elevated lp(a) or PCSK9.

Reviewer 2 Report

No further question

Author Response

We sincerely thank for the endorsement of our response by the reviewer.

Reviewer 4 Report

1. As referred in the previous report, determinations presented at Table 1 that were not referred at Material and Methods (as blood cell count, creatinine, glucose, HbA1c, hs-CRP, cTNI, NT-proBNP), neither the methods used for their evaluation, should be referred in the manuscript. It´s inappropriate to present values of analytical determinations that were not mentioned in the section Material and Methods.

2. At Results, it was stated “Interactions …but were attenuated between LL-37 and PCSK9 (P interaction = 0.094). according to P-value, statistical significance was lost, not attenuated.

3. Titles of figures 4, 5, 6, 7, 8 and 9 could be improved [for instance, for figure 1, replaced by “Associations between LL-37 and the risk of MACE attending to lipid profile values”]. In the legends the definition of MI and, in some cases, of BP must be included.

Author Response

  1. As referred in the previous report, determinations presented at Table 1 that were not referred at Material and Methods (as blood cell count, creatinine, glucose, HbA1c, hs-CRP, cTNI, NT-proBNP), neither the methods used for their evaluation, should be referred in the manuscript. It´s inappropriate to present values of analytical determinations that were not mentioned in the section Material and Methods.

Author reply:

As suggested by the reviewer, we have added details for determinations of blood cell count, creatinine, glucose, HbA1c, hsCRP, cTnI and NT-proBNP, and have rewritten relevant contents in the section of Material and Methods.

(Page 2, line 86-95) Blood samples for complete blood count, basic metabolic panel (e.g., creatinine, glucose), cardiac troponin I (cTnI) and N-terminal prohormone of brain natriuretic peptide (NT-proBNP) were immediately collected via cubital vein right after patient admission to the hospital. Blood cell count was carried out using automatic hematology analyzer (XT-1800i; Sysmex Corporation). Concentrations of blood glucose and creatinine were measured with automatic biochemistry analyzer (Hitachi 7150, Japan) following standard protocols set up by the core laboratory. Serum cTnI was determined using the immunochemiluminometric assay (Access AccuTnI, Beckman Coulter, California). Plasma levels of NT-proBNP were measured with electro-chemiluminescent immunoassays (Elecsys proBNP II assay, Roche Diagnostics, Mannheim, Germany).

(Page 3, line 103-110) Blood samples for lipid profiles and other routine tests were collected via cubital vein … HbA1c levels were measured using Tosoh Automated Glycohemoglobin Analyzer (HLC-723G8, Tokyo, Japan). The level of high-sensitivity C-reactive protein was measured using immunoturbidimetry (Beckmann Assay, Bera, California)

  1. At Results, it was stated “Interactions …but were attenuated between LL-37 and PCSK9 (P interaction = 0.094). according to P-value, statistical significance was lost, not attenuated.

Author reply:

As suggested by the reviewer, we have modified the relevant sentence:

(Page 9, line 235-236) Interactions remained significant between LL-37 and lp(a) (P interaction = 0.036), but were not statistically significant between LL-37 and PCSK9 (P interaction = 0.094).

  1. Titles of figures 4, 5, 6, 7, 8 and 9 could be improved [for instance, for figure 1, replaced by “Associations between LL-37 and the risk of MACE attending to lipid profile values”]. In the legends the definition of MI and, in some cases, of BP must be included.

Author reply:

Thanks for the suggestions and careful considerations by the reviewer. We have accordingly modified the titles of relevant figures and checked the legends.

(Page 7, line 212) Figure 4. Associations between LL-37 and the risk of MACE according to lipid profile values.

(Page 8, line 241-247) Figure 5. Associations between LL-37 and the risk of secondary composite endpoint (cardiac death, recurrent MI and ischemic stroke) according to lipid profile values … MI = myocardial infarction …

(Page 9, line 255-261) Figure 6. Associations between LL-37 and the risk of all-cause death according to lipid profile values … BP = blood pressure …

(Page 10, line 271-277) Figure 7. Associations between LL-37 and the risk of cardiac death according to lipid profile values … BP = blood pressure …

(Page 10, line 280-284) Figure 8. Associations between LL-37 and the risk of recurrent MI according to lipid profile values … MI = myocardial infarction …

(Page 11, line 287-292) Figure 9. Associations between LL-37 and the risk of ischemic stroke according to lipid profile values … BP = blood pressure …